# EGOEXO-CON: EXPLORING VIEW-INVARIANT VIDEO TEMPORAL UNDERSTANDING

## ABSTRACT

Can Video-LLMs achieve *consistent* temporal understanding when videos capture the same event from different viewpoints? To study this, we introduce EgoExo-Con (Consistency), a benchmark of comprehensively synchronized egocentric and exocentric video pairs with human-refined queries in natural language. EgoExo-Con emphasizes two temporal understanding tasks: Temporal Verification and Temporal Grounding. It evaluates not only correctness but consistency across viewpoints. Our analysis reveals two critical limitations of existing Video-LLMs: (1) models often fail to maintain consistency, with results far worse than their single-view performances. (2) When naively finetuned with synchronized videos of both viewpoints, the models show improved consistency but often underperform those trained on a single view. For improvements, we propose View-GRPO, a novel reinforcement learning framework that effectively strengthens view-specific temporal reasoning while encouraging consistent comprehension across viewpoints. Our method demonstrates its superiority over naive SFT and GRPO, especially for improving cross-view consistency. All resources will be made publicly available.

## 1 INTRODUCTION

Recent advances in video large language models (Video-LLMs) have shown impressive capabilities in question answering (Zhang et al., 2024b; Wang et al., 2024a; 2025a) and temporal grounding (Ren et al., 2024; Huang et al., 2024; Zeng et al., 2024; Wang et al., 2025b), and are stepping towards fine-grained and long-ranged reasoning (Zhang et al., 2024a; Shen et al., 2024; Wang et al., 2025c; Feng et al., 2025a). However, most benchmarks (Mangalam et al., 2023; Fu et al., 2024; Wu et al., 2024; Zhou et al., 2025) and methods assume a fixed or minimally varying viewpoint, *e.g.* third-person view (exo) videos, leaving open a critical question: *can Video-LLMs achieve consistent temporal understanding across different camera perspectives?*

Often, videos of the same event appear strikingly different when captured from different perspectives. A cooking demonstration filmed from a head-mounted camera (ego view) looks unlike a side-mounted tripod shot (exo view). Yet, the underlying temporal dynamics, such as cutting vegetables and stirring a pot, are identical. For humans, this view variation rarely impedes understanding; we easily track the sequence of actions and localize their temporal moments across viewpoints. This makes temporal reasoning particularly critical: while appearance cues can vary drastically with viewpoint, the temporal structure of events is invariant. Thus, evaluation of cross-view consistency is essential and can be effectively carried out through temporal understanding tasks; however, such capabilities remain largely underexplored in current Video-LLMs.

To study this, we introduce EgoExo-Con, a benchmark comprising 491 synchronized ego-exo video pairs and 3,178 temporal-bounded event queries, to evaluate whether models can provide consistent predictions across viewpoints - a key indicator of view-invariant video-language understanding. The benchmark focuses on two temporal understanding tasks: *temporal verification* (Jung et al., 2025) and *temporal grounding* (Gao et al., 2017). Temporal verification is a binary QA task that asks whether a given event occurs within a specific video moment, while temporal grounding requires identifying the relevant video moment (start and end timestamps) corresponding to an event query. In both tasks, we ask the same event but with synchronized videos of different viewpoints, and check if the tested models can output correct and consistent responses.

We evaluate both the advanced closed-source models (Comanici et al., 2025; OpenAI, 2025) and open-source Video-LLMs comprising general-purpose (Li et al., 2024a; Zhang et al., 2025a; Bai et al., 2025; Cheng et al., 2024b) and time-aware variants (Ren et al., 2024; Huang et al., 2024; Zeng et al., 2024; Jung et al., 2025). Our benchmark results reveal that all models, especially the open-source ones, struggle with cross-view consistency. They generally exhibit a modest performance gap between individual ego and exo videos, but achieve consistency scores barely over half their single-view performance in both tasks. This indicates that the relatively stable performances across viewpoints may be sourced from view-specific biases rather than robust cross-view temporal understanding. Our further investigations show that naive multi-view supervised fine-tuning (SFT) with synchronized video-language data is insufficient. In fact, it even underperforms the counterpart with single-view training, reflecting that naively merging viewpoints may introduce conflicting priors, undermining temporal signals and consistency. These collectively suggest that *viewpoint variation remains a significant challenge for current Video-LLMs in robust video temporal understanding*.

Finally, we propose View-GRPO, a reinforcement learning (RL) framework that strengthens temporal reasoning across viewpoints while aligning the final conclusions. Experiments demonstrate that our method yields more robust and consistent video understanding than standard finetuning. In summary, EgoExo-Con establishes a new paradigm for evaluating and improving view-invariant temporal understanding in Video-LLMs. We hope it will foster future research on models that truly capture the essence of dynamic events independent of perspective. Our primary contributions are as follows:

- We study the robustness of Video-LLMs in cross-view video temporal understanding and introduce EgoExo-Con, a synchronized ego–exo benchmark constructed with manual annotation efforts.

- We reveal that current Video-LLMs achieve cross-view consistency barely better than half of their sing-view performance, and naively blending perspectives for training could introduce conflicting priors, undermining consistency rather than improving it.

- We propose View-GRPO and construct View30K, a reinforced approach and dataset to explicitly strengthen temporal reasoning while encouraging view-invariant comprehension, significantly outperforming naive SFT and GRPO.

## 2 RELATED WORK

**Video Large Language Models.** Video-LLMs (Li et al., 2024a; Zhang et al., 2024b; 2025a; Bai et al., 2025; Wang et al., 2025a; Comanici et al., 2025) integrate pretrained video representations into powerful LLMs (Grattafiori et al., 2024; Bai et al., 2025; Yang et al., 2025) to enable chatting about videos. While advances to date are mostly achieved in short and coarse-grained question answering (Xu et al., 2017; Yu et al., 2019), more recent Video-LLMs (Ren et al., 2024; Huang et al., 2024; Qian et al., 2024; Guo et al., 2024; Wang et al., 2024b; Zeng et al., 2024; Meinardus et al., 2024; Jung et al., 2025; Li et al., 2025) have explored grasping fine-grained temporal moments ("when"). However, all of these models solve videos captured in a single camera viewpoint (either exo or ego) and do not evaluate whether temporal reasoning remains stable across views of the same event. This work thus fills such gap by conducting a comprehensive analysis.

**Ego-Exo Benchmarks.** Most benchmarks target either exocentric (Gao et al., 2017; Yu et al., 2019; Xiao et al., 2021; 2024; Fu et al., 2024) or egocentric (Mangalam et al., 2023; Di & Xie, 2024; Cheng et al., 2024a; Ye et al., 2015; Xiao et al., 2025) video understanding, with only a few offering paired views: CharadesEgo (Sigurdsson et al., 2018), LEMMA (Jia et al., 2020), EgoExo-4D (Grauman et al., 2024a), Assembly101 (Sener et al., 2022), EgoExo-Fitness (Li et al., 2024b), and EgoExOR (Özsoy et al., 2025). Yet, they are either domain-specific or do not evaluate cross-view temporal reasoning. A concurrent effort, EgoExoBench (He et al., 2025), also explores Video-LMMs in cross-view temporal reasoning, but it primarily targets action ordering via multi-choice selection, and does not consider prediction consistency. Our EgoExo-Con, however, studies a more challenging setting in aligning event queries with corresponding local video moments across views, and highlights cross-view consistency.

**Ego-Exo Learning.** Research on egocentric–exocentric video understanding primarily studies representation alignment and cross-view adaptation. For instance, prior efforts (Sigurdsson et al., 2018; Wang et al., 2023; Xue & Grauman, 2023; Luo et al., 2025) in action recognition have proposed self-supervised methods based on contrastive objectives for view-invariant representation learning.

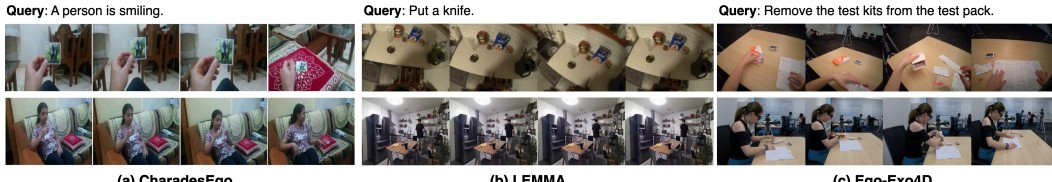

Figure 1: **Examples of queries and corresponding video moments from existing datasets.** (a) and (b) highlight fundamental limitations, with the egocentric view (top) in (a) being insufficient due to differing focuses, and the exocentric view (bottom) in (b) being ambiguous due to occlusion and distance. Although the query in (c) is identifiable from both viewpoints, we enrich it with details.

Another line of studies (Li et al., 2021; Luo et al., 2024; Zhang et al., 2025b) has explored distilling knowledge from one view to another. However, these efforts rarely examine whether the temporal reasoning of Video-LLMs remains consistent when identical events are observed from different viewpoints. In this work, we investigate this and propose a RL-based approach to improve temporal reasoning consistency across viewpoints, inspired by the recent success of RL learning in video reasoning (Feng et al., 2025b; Wang et al., 2025b; Liao et al., 2025; Zhang et al., 2025c).

## 3 EGOEXO-CON DATASET

### 3.1 DATA COLLECTION

We source data from three datasets: CharadesEgo (Sigurdsson et al., 2018), LEMMA (Jia et al., 2020), and EgoExo-4D (Grauman et al., 2024b), as they cover diverse and general domains, whereas other benchmarks are often restricted to specific domains (*e.g.*, fitness (Li et al., 2024b), toy assembly (Sener et al., 2022), and surgery (Özsoy et al., 2025)). CharadesEgo and LEMMA feature daily human-object interactions, while EgoExo-4D spans diverse skilled tasks, such as bike repairing and rock climbing. Among them, we collect synchronized video data annotated with query-timestamp pairs to support our focus on temporal understanding tasks. To ensure reliable evaluation, we carefully balance video diversity with model feasibility. For example, our preliminary experiments find that current models hardly perform effective temporal localization for long videos, making it infeasible to analyze consistency. Thus, we segment videos longer than five minutes into multiple clips surrounding the ground-truth moments, with each video clip lasting for at least two minutes, thus maximally preserving content diversity while keeping the task manageable for existing Video-LLMs.

### 3.2 DATA FILTERING AND REFINEMENT

Unfortunately, the original temporal queries in datasets often do not meet our requirements. In Fig. 1, (a) queries in CharadesEgo are template-based, with categorical actions, and (b) queries in LEMMA rely on atomic actions and objects, both of which tend to miss details. More critically, viewpoint-induced ambiguities hinder reliable evaluation for cross-view consistency: key elements may be visible from one viewpoint but obscured from another due to varied focus and temporal alignment. For instance, the query "A person is smiling" in Fig. 1-(a) is not visible from the egocentric video. Overlooking these issues undermines reliable evaluation and analysis.

To address this, we reformulate queries in multiple stages. We convert the per-frame Human-Object Interaction (HOI) labels (*e.g.*, put + cup, fridge) in LEMMA into natural-language queries. Since HOI labels are grounded to short 1–2 second intervals, we aggregate consecutive annotations into longer spans, extract salient verbs and nouns, and verbalize them into natural-language queries using simple rules for targets and prepositions. Next, we utilize a powerful large model (*i.e.*, GPT-4o (Achiam et al., 2023)) to enrich the original queries across all datasets. Specifically, given sampled frames from the target moments, the model verifies whether a query contains elements that cannot be reliably inferred from one or both viewpoints and produces refined alternatives. Additionally, the model generates a *misaligned query* containing irrelevant content, which serves as a negative sample for temporal verification, thus balancing answers for "yes" and "No" in verification task. The full prompt is shown in Appendix Fig. 11.

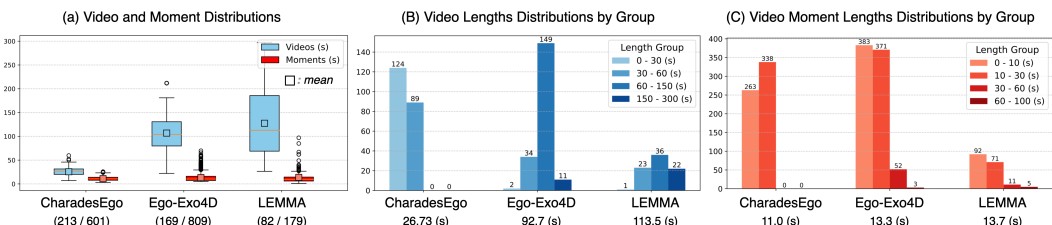

Figure 2: **Statistics of EgoExo-Con.** The numbers below (a) show the video and moment counts per subset, and those in (b) and (c) show their average lengths, respectively. The statistics suggest the high diversity of EgoExo-Con in data sources, video and moment lengths.

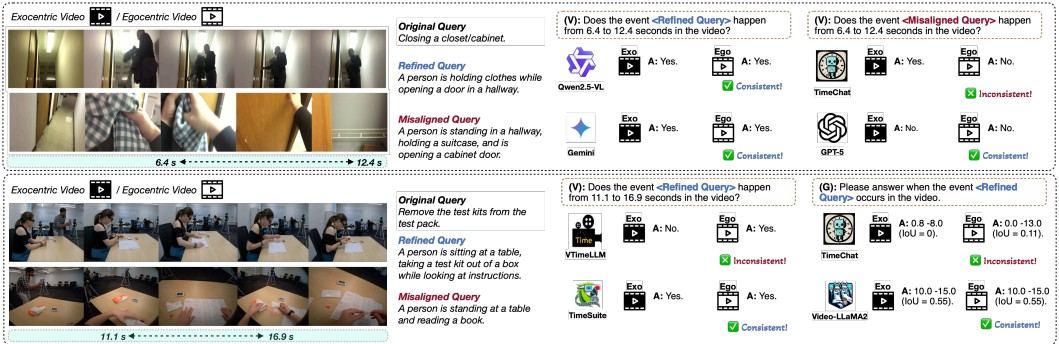

Figure 3: **Examples of test data and the corresponding model responses.** We create refined and misaligned queries from each original query, use them for temporal verification (V) and grounding (G), and assess cross-view answer consistency.

Finally, we perform human validation for all samples. Four human evaluators review the generated queries alongside associated synchronized video pairs. They confirm whether refined queries are accurately grounded in both viewpoints, and misaligned queries intentionally conflict with the visual content. Queries passing this validation are retained, while ambiguous or low-quality samples are refined further or discarded (*e.g.*, if the video itself is too noisy). Uncertain cases are cross-checked with the authors to ensure reliability. Fig. 3 presents examples of refined and misaligned queries generated from the original query and its associated video.

### 3.3 DATASET STATISTICS

Eventually, we obtain 491 synchronized video pairs (213, 169, and 82 pairs are from CharadesEgo, EgoExo4D, and LEMMA, respectively) and 3,178 tightly aligned queries with timestamps. As shown in Fig. 2, each subset introduces distinct challenges specific to its domain and contributes a diverse range of video and moment lengths. Collectively, they provide extensive coverage of content, making EgoExo-Con a highly diverse and comprehensive benchmark to challenge temporal capabilities.

## 4 EVALUATION

### 4.1 MODELS

**Baseline.** We evaluate a series of open-sourced models and categorize them as general-purpose or time-aware models, depending on whether they are designed for generic question-answering or specifically tuned to provide answers with corresponding video timestamps. Four general-purpose models: VideoChat2 (Li et al., 2024a), Qwen2.5-VL (Bai et al., 2025), and Video-LLaMA2 (Cheng et al., 2024b), and Video-LLaMA3 (Zhang et al., 2025a), and four time-aware models: VTimeLLM (Huang et al., 2024), TimeChat (Ren et al., 2024), TimeSuite (Zeng et al., 2024), and TimeChat-VT (Jung et al., 2025), are included. We provide details of each model in Appendix A. Additionally, we include two powerful closed-source models: GPT-5 (OpenAI, 2025) and Gemini-2.5 Flash (Comanici et al., 2025). We also benchmark human performance as a reference. We invite four evaluators and present

Table 1: **Performance on EgoExo-Con.** F: input frames. Ego: include ego data for training.

| Methods | # F | Ego | EgoExo-Con | | | | | |
|---|---|---|---|---|---|---|---|---|
| | | | V-Exo | V-Ego | V-ExoEgo | G-Exo | G-Ego | G-ExoEgo |
| Human | - | - | 92.1 | 91.3 | 89.4 | 72.4 | 73.0 | 67.3 |
| *Closed-source* | | | | | | | | |
| GPT-5 (OpenAI, 2025) | 32 | - | 60.5 | 61.3 | 52.5 | 34.5 | 32.8 | 20.1 |
| Gemini-2.5 Flash (Comanici et al., 2025) | 1 fps | - | 70.4 | 70.1 | 52.3 | 42.0 | 45.9 | 20.8 |
| Random | - | - | 50.0 | 50.0 | 50.0 | 12.5 | 12.5 | 12.5 |
| *General-purpose* | | | | | | | | |
| VideoChat2 (Li et al., 2024a) | 16 | ✓ | 46.0 | 45.1 | 23.4 | 5.6 | 5.3 | 4.0 |
| Qwen2.5-VL (Bai et al., 2025) | 1 fps | ✗ | 54.3 | 56.3 | 33.0 | 14.2 | 11.4 | 6.9 |
| Video-LLaMA2 (Cheng et al., 2024b) | 8 | ✓ | 53.3 | 52.1 | 27.9 | 12.0 | 11.5 | 7.5 |
| Video-LLaMA3 (Zhang et al., 2025a) | 1 fps | ✓ | 56.7 | 54.6 | 36.6 | 27.7 | **28.0** | 16.2 |
| *Time-aware* | | | | | | | | |
| VTimeLLM (Huang et al., 2024) | 100 | ✗ | 48.9 | 48.5 | 23.5 | 12.6 | 11.1 | 6.5 |
| TimeChat (Ren et al., 2024) | 96 | ✗ | 48.9 | 48.4 | 25.1 | 21.3 | 20.5 | 12.8 |
| TimeSuite (Zeng et al., 2024) | 128 | ✓ | 47.4 | 48.5 | 25.6 | **28.2** | 27.3 | **18.7** |
| TimeChat-VT (Jung et al., 2025) | 96 | ✗ | **62.1** | **61.4** | **42.1** | 27.8 | 26.2 | 16.3 |

each viewpoint independently to avoid biased predictions, reporting the average of their scores. Note that closed-source models and human performance are reported on a ≃30% subset of the full benchmark, which was uniformly sampled from each split to control evaluation costs. Additionally, we benchmark a random method that always returns "yes" or the entire video span for temporal verification and grounding, respectively.

**Evaluation Metric.** We use accuracy in percentage for temporal verification (V) and R@1, Intersection over Union (IoU)=0.5 for temporal grounding (G). For grounding, predictions are considered correct if their IoU with the ground-truth moment exceeds 0.5. A model is evaluated separately for each viewpoint: V-Ego and V-Exo measure binary accuracy for egocentric and exocentric videos, respectively. Similarly, G-Ego and G-Exo denote grounding performance. Consistency metrics, V-EgoExo and G-EgoExo, measure whether a model correctly verifies or grounds specific moments (*i.e.*, IoU < 0.5) for both synchronized videos; consistent but wrong answers are not considered.

## 4.2 PERFORMANCE ANALYSIS ON EGOEXO-CON

Our analyses on the results of EgoExo-Con in Table 1 are as follows: **(1) Single view vs. Cross view.** While all models show a modest performance gap between individual ego and exo videos, they struggle with cross-view consistency in both tasks. Open-source models in particular achieve barely half of their single-view performance. This demonstrates that the relatively stable performances across viewpoints are largely due to view-specific bias cues but not robust cross-view reasoning. **(2) Time-aware vs. General-purpose.** Time-aware models generally lead on grounding. TimeSuite attains the strongest grounding consistency, and TimeChat-VT is competitive while also giving the best verification consistency. This advantage likely comes from better instruction tuning. However, VTimeLLM underperforms most general-purpose models, showcasing that time-aware models are not necessarily superior to general-purpose ones in temporal reasoning. **(3) Closed-sourced vs Human.** Closed-source models generally outperform open-source ones, reflecting their stronger capabilities. Yet a substantial gap (37% 47%) in cross-view consistency remains compared to humans, with scores approaching random, underscoring the challenge of EgoExo-Con and the room for improvement. **(4) Training with ego data is not sufficient.** Models including egocentric videos for training do not consistently yield higher consistency than others trained on exocentric videos alone, showing that simply mixing ego and exo data does not benefit consistency. **(5) Temporal reasoning outweighs increasing frames for better results.** Video-LLaMA2 (8 frames) outperforms VideoChat2 (16 frames) across all metrics. TimeChat-VT (96 frames) also outperforms several models that use more/less context, suggesting reasoning and temporal modeling matter more than sheer frame count.

Furthermore, we analyze model behaviors across subsets by plotting the ego–exo performance gap in Fig. 4. Patterns are consistent across grounding and verification: in CharadesEgo, most models perform better on exocentric views (blue), whereas in LEMMA, they tend to favor egocentric views (red); EgoExo-4D shows mixed but generally smaller gaps. These trends likely reflect domain characteristics. As shown in Fig. 3, when a person performs in a fixed position, egocentric videos

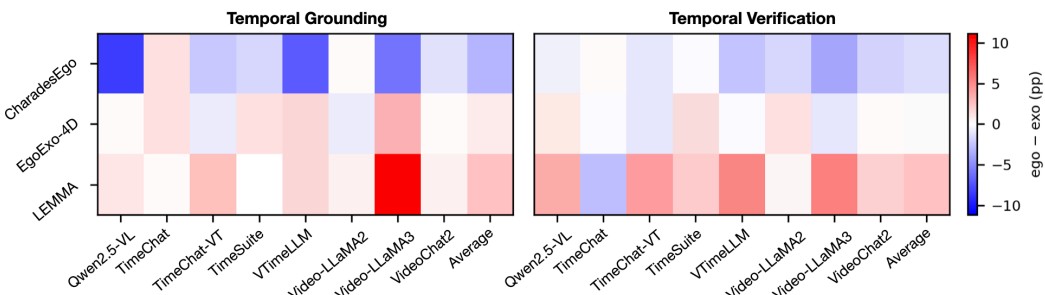

Figure 4: **Heatmaps of the performance gap.** All values are reported in percentage points. Red and blue indicate higher performances on *ego* and *exo* perspectives, respectively. *i.e., a blue cell indicates that the corresponding model performs better on exocentric videos than on egocentric ones.*

Table 2: **Fine-tuned performance.** The left and right tables report model performance for temporal verification (V) and temporal grounding (G). The first row in each model reports zero-shot performance, while the subsequent rows present results after fine-tuning on either (+Ego, +Exo) or both viewpoints (+ EgoExo). Notably, models trained on both viewpoints are not always the best-performing, and then they barely outperform those trained on a single viewpoint.

| Methods | CharadesEgo | | | EgoExo-4D | | |
|---|---|---|---|---|---|---|
| | V-Exo | V-Ego | V-ExoEgo | V-Exo | V-Ego | V-ExoEgo |
| VideoChat2 | 46.3 | 44.4 | 22.2 | 41.4 | 41.5 | 19.8 |
| + Ego | 56.2 | **59.2** | **36.4** | 46.5 | 49.2 | 29.7 |
| + Exo | **56.6** | 57.5 | 35.5 | 46.1 | 47.3 | 29.4 |
| + EgoExo | 56.4 | 57.1 | 34.7 | **48.3** | **50.1** | **30.1** |
| Video-LLaMA2 | 54.0 | 52.4 | 28.2 | 51.5 | 52.8 | 28.1 |
| + Ego | 57.0 | **58.2** | **31.7** | **60.2** | **60.3** | 38.8 |
| + Exo | 57.6 | 56.1 | 31.4 | 59.8 | 60.1 | **39.2** |
| + EgoExo | **58.5** | 57.3 | 31.0 | 57.5 | 59.6 | 39.1 |

| Methods | CharadesEgo | | | EgoExo-4D | | |
|---|---|---|---|---|---|---|
| | G-Exo | G-Ego | G-ExoEgo | G-Exo | G-Ego | G-ExoEgo |
| TimeChat | 44.9 | 46.1 | 30.1 | 4.9 | 6.3 | 3.3 |
| + Ego | **62.0** | **62.1** | **48.3** | 9.7 | **13.1** | **4.9** |
| + Exo | 58.8 | 60.0 | 47.1 | 10.4 | 10.7 | 4.2 |
| + EgoExo | 60.3 | 61.8 | 46.5 | **10.6** | 11.5 | 4.3 |
| TimeSuite | 63.4 | 56.2 | 44.8 | 5.8 | 8.3 | 2.3 |
| + Ego | 61.0 | 61.5 | 54.0 | 6.7 | 6.2 | 4.8 |
| + Exo | **74.6** | **68.7** | **59.5** | **10.5** | **9.9** | **5.7** |
| + EgoExo | 67.8 | 61.1 | 51.3 | 9.6 | 8.9 | 5.3 |

often provide favorable cues such as clearer hand–object interactions. Conversely, when a person moves around or changes location, egocentric views can become more challenging due to rapid scene shifts, while exocentric views offer greater stability. Such trends are particularly pronounced in CharadesEgo. Detailed results across subsets are provided in Appendix Table 5. Although the relative effectiveness of each viewpoint varies by domain, they do not significantly affect overall consistency. Overall, across models, consistency scores lag far behind single-view metrics, underscoring that achieving robust, view-invariant temporal understanding remains an open challenge.

## 4.3 SUPERVISED PERFORMANCE ON EGOEXO-CON

In this section, we study whether supervised fine-tuning (SFT) with synchronized ego-exo video data improves performance. We first collect 3.6k from CharadesEgo and 2.3k videos from EgoExo-4D training sets. We exclude LEMMA due to its limited size (*i.e.*, 243 videos for training). Then we finetune two general-purpose models: VideoChat2 and Video-LLaMA for temporal verification, and two time-aware models: TimeChat and TimeSuite for temporal grounding. All models apply LoRA (Hu et al., 2022) fine-tuning with their official configurations. More implementation details are presented in Appendix B.2.

Table 2 reports results of three different settings: training on either viewpoint (Ego, Exo) or both viewpoints together (ExoEgo). All of them consistently improve over zero-shot baselines. Interestingly, despite utilizing twice the data, training on both viewpoints (ExoEgo) yields only marginal gains and often underperforms the models trained on a single view. Specifically, in CharadesEgo, TimeSuite shows a notable 8.1% gap in consistency between training on both viewpoints and training only on exocentric videos. Furthermore, the improvements are still limited when unfreezing the visual encoder (refer to Appendix Table 4).

The above findings are consistent with observations in cross-view learning (Li et al., 2024b; Sigurdsson et al., 2018), where naively blending perspectives does not always bring improvement. Without explicit alignment, conflicting priors across tasks or domains undermine temporal signals and consistency rather than improvement.

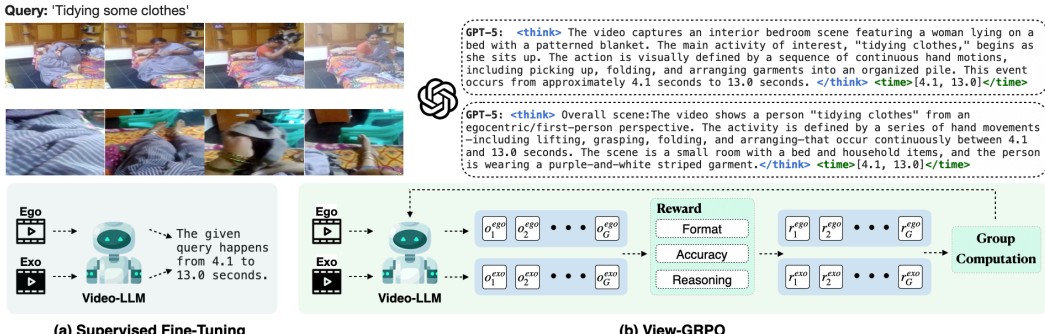

Figure 5: **Overview of our approach.** (a) In supervised fine-tuning, the model is trained to directly predict the same query answers (*e.g.*, video moments) for synchronized video pairs. (b) View-GRPO trains a model to provide viewpoint-specific reasoning chains, which are generated by GPT-5 (top).

## 5 METHOD

Although synchronized videos depict the same content, the reasoning process often differs across viewpoints because of distinct focuses and perspectives. To address this, we propose a reinforcement learning (RL) framework that guides models toward developing viewpoint-specific reasoning while encouraging shared consistency. Rather than simply enforcing identical outputs, our approach explicitly promotes robust reasoning across viewpoints. We build on Group Relative Policy Optimization (GRPO), which is particularly well-suited as it leverages relative rewards instead of absolute scores.

### 5.1 BACKGROUND: GROUP RELATIVE POLICY OPTIMIZATION (GRPO)

GRPO (Shao et al., 2024) is a reinforcement learning algorithm designed to refine large language model outputs by leveraging relative ranking among multiple candidate responses. Instead of treating each response independently with absolute rewards, GRPO evaluates a set of responses produced for the same prompt, assigning rewards in relation to the group. This group-wise normalization helps reduce reward variance and makes optimization more stable compared to approaches relying solely on absolute scores or pairwise comparisons.

Given a prompt $p$, the model generates $G$ candidate responses $o = \{o_1, \ldots, o_G\}$. Each response receives a reward value $r(o_i)$. GRPO then standardizes these scores within the group and optimizes a weighted objective:

$$R(o) = \sum_{i=1}^{G} \frac{\pi_\theta(o_i)}{\pi_{\theta_{old}}(o_i)} \cdot \frac{r(o_i) - \text{mean}(\{r(o_i)\}_{i=1}^{G})}{\text{std}(\{r(o_i)\}_{i=1}^{G})}, \tag{1}$$

where $\pi_\theta(o)$ denotes the current policy and $\pi_{\theta_{old}}(o)$ is the previous policy. To prevent divergence from the base model, KL-divergence regularization is added:

$$\max_{\pi_\theta} \ \mathbb{E}_{o \sim \pi_{\theta_{old}}(p)} \Big[ R(o) - \beta D_{KL}\big(\pi_\theta \parallel \pi_{ref}\big) \Big], \tag{2}$$

where $\pi_{ref}$ is the base model and $\beta$ controls the strength of the regularization. Please refer to the original paper for more details.

### 5.2 LEARNING TEMPORAL REASONING ACROSS VIEWPOINTS

To adapt GRPO for cross-view reasoning, we first curate training data, including temporal reasoning chains for both egocentric and exocentric views. Fig. 5 (Top) illustrates the generation of video reasoning data. We prompt GPT-5 to produce step-by-step reasoning chains for each video in order to solve the given task, ensuring viewpoint-specific reasoning while aligning the final answers. We discard samples lacking valid reasoning in either view, as these typically indicate ambiguous or low-quality data. Specifically, we exclude cases where the model explicitly states its failure in the

Table 3: **Performance of View-GRPO on EgoExo-Con.**

| Methods | EgoExo-Con | | | | | |
|---|---|---|---|---|---|---|
| | V-Exo | V-Ego | V-ExoEgo | G-Exo | G-Ego | G-ExoEgo |
| Qwen2.5-VL-3B | 51.0 | 52.5 | 28.1 | 10.1 | 9.9 | 7.9 |
| + SFT | 52.7 | 51.2 | 30.6 | 16.3 | 16.6 | 12.9 |
| + GRPO | 52.5 | 52.9 | 30.3 | 15.4 | 16.3 | 10.3 |
| **+ View-GRPO** | **54.0** ↑**3.0** | **54.1** ↑**1.6** | **33.9** ↑**5.8** | **18.6** ↑**8.5** | **17.9** ↑**8.0** | **14.8** ↑**6.9** |
| Qwen2.5-VL-7B | 54.3 | 56.3 | 33.0 | 14.2 | 11.4 | 6.9 |
| + SFT | 57.6 | 58.0 | 41.4 | 18.3 | 17.8 | 14.9 |
| + GRPO | 55.2 | 57.6 | 39.8 | 18.6 | 16.1 | 14.3 |
| **+ View-GRPO** | **58.3** ↑**4.0** | **58.1** ↑**1.8** | **44.7** ↑**11.7** | **21.5** ↑**7.3** | **21.0** ↑**9.6** | **18.3** ↑**11.4** |

answer, or where the predicted moment has a temporal IoU (tIoU) below 0.7 with the ground-truth. After filtering, we retain 3.3k videos with 30k reasoning instances, which we name as View30K.

With the dataset, we then design reward functions comprising three major components:

**1. Format Reward.** To encourage structured reasoning and easy answer extraction, responses must follow the template: `<think>...</think><answer>...</answer>`. Formally:

$$r_{\text{form}}(o) = \begin{cases} 1, & \text{if } o \text{ follows the required format,} \\ 0, & \text{otherwise.} \end{cases} \quad (3)$$

**2. Accuracy Reward.** We unify task-specific accuracy into $r_{\text{acc}}$. For temporal grounding, the reward is the temporal Intersection-over-Union (tIoU) between ground truth $[t_s, t_e]$ and prediction $[t_s', t_e']$. For verification, it is binary correctness:

$$r_{\text{acc}}(o) = \begin{cases} \frac{|[t_s,t_e] \cap [t_s',t_e']|}{|[t_s,t_e] \cup [t_s',t_e']|}, & \text{if } grounding, \\ \mathbb{1}[o = o^*], & \text{if } verification. \end{cases} \quad (4)$$

**3. Reasoning Reward.** We design $r_{\text{sim}}$ to measure how closely the generated reasoning aligns with the target ones. We employ LLM (*i.e.*, Qwen2.5-3B (Qwen et al., 2024)) as a judge and design a tailored prompt (see Appendix Fig. 12). Specifically, the judge model provides a similarity score on a scale of 0 to 1 between the candidate output $o$ and reference $o^*$.

Overall, the total reward integrates all components to balance correctness and reasoning quality:

$$r(o) = r_{\text{acc}}(o) + r_{\text{form}}(o) + r_{\text{sim}}(o), \quad (5)$$

thus enabling models to learn correct and consistent cross-view temporal reasoning. We name the overall approach View-GRPO, as shown in Fig. 5-(b) (Bottom).

## 5.3 IMPLEMENTATIONS

We use Qwen2.5-VL (Bai et al., 2025) as a base model. During training, all experiments set the same frame sampling rate (*i.e.*, 2 FPS) and freeze the visual encoder and update only the parameters of the LLM. The training involves 8 × A100 GPUs and requires over 1 day for the 3B model and 2 days for the 7B model. Further implementation details are in Appendix B.3.

## 5.4 ANALYSES ON VIEW-GRPO

Table 3 shows the performance of our method, View-GRPO, on EgoExo-Con compared to SFT and basic GRPO implementation with only format and accuracy rewards. View-GRPO, which highlights an additional reasoning reward and related data for GRPO training, consistently outperforms SFT and basic GRPO. The most significant improvements are often on cross-view consistency (V-ExoEgo and G-ExoEgo), although it also benefits individual views. We conjecture that the reasoning reward plays a central role, as it delivers noticeably higher consistency compared to naive GRPO. By encouraging

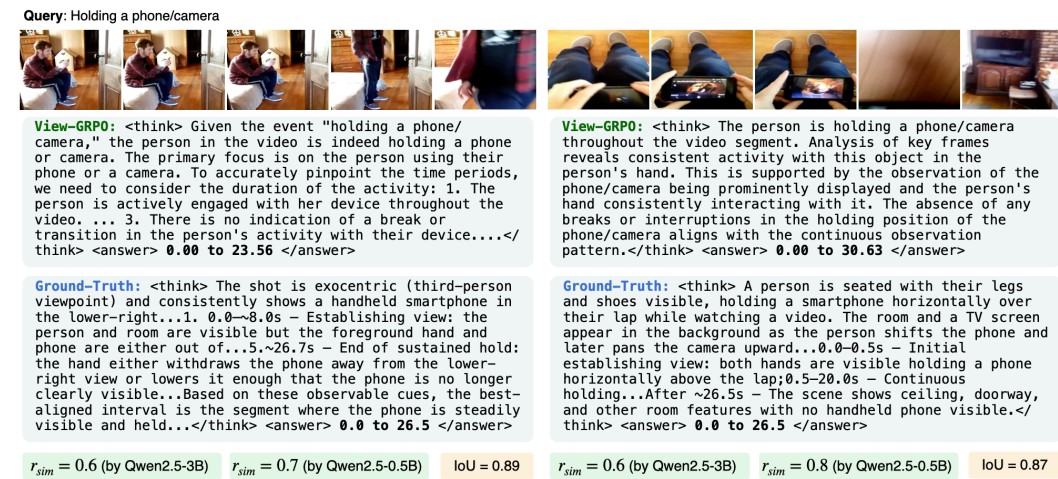

Figure 6: **Visualization of View-GRPO.** The model provides step-by-step temporal reasoning with accurate grounding prediction, achieving high reasoning reward scores from LLM-judges.

models to produce faithful, step-by-step temporal explanations tailored to each viewpoint while converging toward consistent temporal conclusions, the model reduces reliance on view-specific biases and instead learns shared temporal abstractions. Fig. 6 shows the step-by-step temporal reasoning contents generated by View-GRPO and how reliable reasoning gives rise to robust temporal grounding across viewpoints. We also experiment with a different backbone (*i.e.*, InternVL-3.5 (Wang et al., 2025a)) and evaluate on additional benchmarks, Video-MME (Fu et al., 2024) and TVGBench (Wang et al., 2025b), as reported in Appendix Table 6 and 7. Overall, View-GRPO demonstrates its effectiveness and potential for consistent and view-invariant temporal understanding.

While LLMs are commonly used as judges (Zheng et al., 2023; Xie et al., 2023), the role in optimizing models in View-GRPO remains underexplored despite the effectiveness, as they often introduce potential bias and uncertainty in video evaluation (Cores et al., 2024; Liu & Zhang, 2025). To provide insights into this, we employ judge models of different scales (*i.e.*, Qwen2.5-0.5B and Qwen2.5-3B) for temporal grounding and analyze their impact on optimization. In Fig. 7-(a), format and accuracy rewards remain relatively stable across scales. However, Qwen2.5-0.5B produces overly high

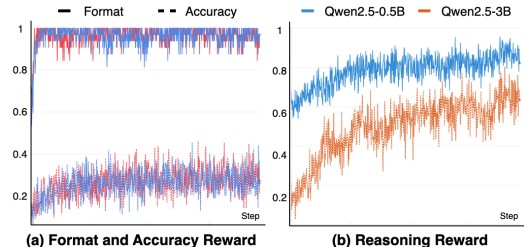

Figure 7: **Reward of different LLM judges.** Qwen2.5-0.5B raises calibration concerns due to its overly high reasoning rewards from early steps.

reasoning rewards from the very first training steps in Fig. 7-(b), raising concerns about calibration and reliability. We find that this behavior leads to a measurable consistency degradation (*i.e.*, –3% in G-EgoExo). We also explore the effectiveness across different reasoning lengths in Appendix C.3. Overall, such results underscore the critical influence of LLM-judges on model optimization.

# 6 CONCLUSION

In this work, we introduce EgoExo-Con that comprises synchronized egocentric and exocentric videos paired with human-refined queries and evaluate whether models perform consistent temporal understanding across viewpoints. EgoExo-Con reveals that Video-LLMs struggle with cross-view consistency, often lagging behind single-viewpoint performance, and that naively training on both viewpoints does not reliably help. To address this, we propose View-GRPO that encourages viewpoint-specific temporal reasoning while promoting cross-view alignment, demonstrating its effectiveness over alternative training strategies. We hope EgoExo-Con establishes a solid benchmark for faithful temporal understanding across viewpoints, and View-GRPO provides insights toward achieving robust, view-invariant video comprehension.

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

APPENDIX

We provide further details that are not included in the main paper due to the page limitation:

A. **Details of Models**: We explain the models that are utilized in our experiments one by one.

B. **Details of Experiments**: We provide further details of our experiments.

C. **Additional Experiments**: We provide additional experiments.

D. **Additional Visualizations**: We provide additional visual examples of model responses on EgoExo-Con and analyze them.

## A    DETAILS OF MODELS

In this section, we describe eight open-source Video-LLMs: VideoChat2 (Li et al., 2024a), Qwen2.5-VL (Bai et al., 2025), Video-LLaMA2 (Cheng et al., 2024b), Video-LLaMA3 (Zhang et al., 2025a), VTimeLLM (Huang et al., 2024), TimeChat (Ren et al., 2024), TimeSuite (Zeng et al., 2024), and TimeChat-VT (Jung et al., 2025), which are utilized in our evaluation. Note that the size of the models is 7B.

1. **VideoChat2** design three different video instruction tuning stages. Specifically, they align multi-modal inputs in the first stage and then generate captions from various image-text pairs. Finally, they conduct instruction tuning to better align responses with human instructions. VideoChat2 demonstrates significant improvements in video question answering benchmarks in zero-shot settings.

2. **Qwen2.5-VL** Qwen2.5-VL is one of the latest models of Qwen vision-language series. It achieves strong benchmark performances through enhanced visual recognition, precise object localization, robust document parsing, and long-video comprehension.

3. **Video-LLaMA2** is one of the state-of-the-art Video-LLMs, demonstrating superior performances on video question-answering tasks. They seamlessly integrate both visual and audio modalities in videos and propose STC connector to understand spatiotemporal video information.

4. **Video-LLaMA3** is a series of Video-LLaMA family. It emphasizes a vision-centric training paradigm and vision-centric framework design. With curated high-quality image and video data, Video-LLaMA3 achieves compelling performances across diverse image and video understanding benchmarks.

5. **VTimeLLM** aims to include video timestamps along with the answer for human instruction. It designs a three-stage instruction tuning. Initially, it trains a visual projection layer with image-text pairs and then incorporates video datasets with temporal modeling. VTimeLLM devises two types of QA dialogue templates, including single-turn and multi-turn, to prompt questions requiring a comprehensive description of all events and their corresponding timestamps.

6. **TimeChat** is developed to localize and identify specific video moments from a given human instruction. It utilizes a time-aware frame encoder that injects timestamp information into visual features, leveraging Q-Former, and designs a sliding video Q-Former to handle temporal information.

7. **TimeSuite** tackles achieving high performance on both question-answering and grounding for long videos. It argues that previous Video-LLMs struggle to achieve both capabilities and develop a VideoChat-T and a temporal-centric instruction-tuning dataset, TimePro.

8. **TimeChat-VT** is a model that specifically takes into account consistency modeling. Upon TimeChat, it develops a new instruction tuning method, VTune, that converts temporal grounding into a verification process. This requires not only precise temporal grounding, but also confirming the occurring events from specific video moments.

Table 4: **Performance comparison between different training settings for a visual encoder.** In Vis. column, ❄ represents when we freeze the visual encoder, while 🔥 denotes the visual encoder is trainable. All models are trained on both viewpoints. If there is an improvement, we use a blue color; otherwise, we use a red color.

| Methods | Vis. | EgoExo-Charades-Con | | | | | | EgoExo-4D-Con | | | | | |
|---|---|---|---|---|---|---|---|---|---|---|---|---|---|
| | | V-Exo | V-Ego | V-ExoEgo | G-Exo | G-Ego | G-ExoEgo | V-Exo | V-Ego | V-ExoEgo | G-Exo | G-Ego | G-ExoEgo |
| VideoChat2 | ❄ | 56.35 | 57.07 | 34.69 | 28.77 | 27.61 | 22.74 | 59.06 | 60.38 | 37.25 | 4.52 | 4.97 | 2.33 |
| | 🔥 | 55.78 | 57.01 | 32.78 | 29.76 | 30.13 | 23.86 | 58.32 | 54.55 | 31.25 | 2.13 | 2.63 | 1.13 |
| TimeChat | ❄ | 52.49 | 48.59 | 27.91 | 60.30 | 61.79 | 46.51 | 51.64 | 50.31 | 28.33 | 10.57 | 11.48 | 4.33 |
| | 🔥 | 50.76 | 45.11 | 21.64 | 55.87 | 59.76 | 42.57 | 51.97 | 51.34 | 29.09 | 10.32 | 11.76 | 4.23 |

Table 5: **Performance on EgoExo-Con across different subsets.** While performance for temporal verification remains steady across subsets, there is a noticeable performance gap between subsets in temporal grounding, likely due to the different difficulty of tasks.

| Methods | CharadesEgo | | | | | | EgoExo-4D | | | | | | LEMMA | | | | | |
|---|---|---|---|---|---|---|---|---|---|---|---|---|---|---|---|---|---|---|
| | V-Exo | V-Ego | V-ExoEgo | G-Exo | G-Ego | G-ExoEgo | V-Exo | V-Ego | V-ExoEgo | G-Exo | G-Ego | G-ExoEgo | V-Exo | V-Ego | V-ExoEgo | G-Exo | G-Ego | G-ExoEgo |
| *General-purpose* | | | | | | | | | | | | | | | | | | |
| VideoChat2 | 48.6 | 53.6 | 27.2 | 13.6 | 12.5 | 10.6 | 41.4 | 41.5 | 19.8 | 0.9 | 1.0 | 0.4 | 40.5 | 42.5 | 21.2 | 0.6 | 1.1 | 0.6 |
| Qwen2.5-VL | 59.1 | 58.4 | 42.2 | 31.4 | 23.1 | 16.1 | 58.4 | 59.3 | 49.5 | 5.3 | 5.3 | 2.7 | 55.6 | 59.2 | 41.9 | 0.6 | 1.7 | 0.0 |
| Video-LLaMA2 | 54.0 | 52.4 | 28.2 | 27.8 | 27.8 | 17.6 | 51.5 | 52.8 | 28.1 | 2.5 | 1.6 | 1.5 | 50.6 | 50.8 | 25.4 | 2.2 | 2.8 | 1.7 |
| Video-LLaMA3 | 61.8 | 57.9 | 40.3 | 57.1 | 51.1 | 37.3 | 52.6 | 51.5 | 33.6 | 8.7 | 12.0 | 3.1 | 53.9 | 59.5 | 36.3 | 11.7 | 22.9 | 5.6 |
| *Time-aware* | | | | | | | | | | | | | | | | | | |
| VTimeLLM | 49.5 | 47.0 | 23.8 | 19.5 | 12.5 | 7.5 | 49.0 | 49.0 | 23.6 | 8.4 | 10.1 | 6.3 | 46.4 | 51.7 | 23.5 | 10.1 | 11.7 | 6.1 |
| TimeChat | 48.0 | 48.1 | 25.2 | 44.9 | 46.1 | 30.1 | 49.5 | 49.3 | 25.4 | 4.9 | 6.3 | 3.3 | 49.2 | 46.4 | 23.5 | 5.0 | 5.0 | 3.4 |
| TimeSuite | 47.7 | 47.6 | 24.7 | 63.4 | 56.2 | 44.8 | 45.8 | 47.3 | 25.2 | 5.8 | 8.3 | 2.3 | 54.0 | 56.3 | 30.6 | 10.6 | 16.2 | 6.7 |
| TimeChat-VT | 64.6 | 63.7 | 44.8 | 56.6 | 54.2 | 35.8 | 60.4 | 59.4 | 38.8 | 9.3 | 8.4 | 3.3 | 60.9 | 65.1 | 45.5 | 10.1 | 12.8 | 3.9 |

# B DETAILS OF EXPERIMENTS

## B.1 PROMPT TEMPLATES

For temporal verification, we give variations to prompt templates beyond utilizing misaligned queries. Specifically, following the previous work (Jung et al., 2025), we also include templates like "Is the event missing in the video?" or "Is the event not present in the video?" to shift the correct answer "Yes." to "No."

For temporal grounding, some general-purpose models do not officially provide prompt templates for grounding. Therefore, we borrow the prompts from the previous work (Jung et al., 2025) for the general-purpose models and closed-sourced models, and design the prompt *"Give the query, when does the described content occur in the video? Please return its start and end time using 'start - end seconds'."* to ensure the model includes timestamps in its answer.

## B.2 DETAILS OF SUPERVISED FINE-TUNING FOR VIDEO-LLMS

For each task, we design question–answer templates. In temporal verification, we use two formats: "Does 'event' happen from 'st' to 'ed' in the video?" and "Does 'event' not happen from 'st' to 'ed' in the video?". Here, 'event', 'st', and 'ed' are replaced with the annotated query and its start and end timestamps. Answers are restricted to "yes" or "no." For temporal grounding, the template "Localize the 'event' in the video and return its start and end times." is employed. Note that both tasks utilize the same number of videos and queries. Again, we follow the official code and configurations in each model. All experiments run 3 epochs with 4 × A100 GPUs.

## B.3 DETAILS OF VIEW-GRPO TRAINING

To generate View30K, we design prompts in Fig. 13 and 14 for each task. We sample frames every 1 second (*i.e.*, 1 FPS) from videos and give them to GPT-5. We perform batch processing using GPT-API, and it involves less than 1 day for 3.6k videos. Additionally, the model fails to generate reasoning data for 0.3k videos, and a total of 3.3k with 61k reasoning for each task remains. For training, we set the max pixels for video processing as 2.8M. We use the AdamW optimizer with a learning rate of 1e-6 and set the batch size to 8, 1 batch for each GPU.

Table 6: **Performance of InternVL-3.5 on EgoExo-Con.**

| Methods | EgoExo-Con | | | | | |
| --- | --- | --- | --- | --- | --- | --- |
| | V-Exo | V-Ego | V-ExoEgo | G-Exo | G-Ego | G-ExoEgo |
| InternVL3.5-8B | 64.4 | 64.7 | 50.7 | 12.8 | 6.7 | 3.0 |
| + **View-GRPO** | 73.1 | 74.4 | 62.4 | 20.5 | 16.8 | 10.6 |

## C  ADDITIONAL EXPERIMENTS

### C.1  THE IMPACT OF UNFREEZING VIDEO ENCODER

In Table 4, we conduct fine-tuning models: VideoChat2 (Li et al., 2024a) and TimeChat (Ren et al., 2024) when fine-tuned with their video encoders unfrozen. Despite updating the visual encoders during training, we observe no further improvements; in fact, the models often underperform compared to the settings with frozen encoders. We conjecture that this may be due to overfitting or the limited scale of training data, which may not sufficiently support end-to-end tuning of large visual backbones. Furthermore, this supports the previous findings that naively mixing both viewpoints does not easily lead to improved view-invariant understanding.

### C.2  PERFORMANCE ON EGOEXO-CON ACROSS SUBSETS

Table 5 reports performance across models and subsets. The performance for temporal verification remains steady across subsets. In contrast, there is a noticeable performance gap between CharadesEgo and the others. Specifically, the models tend to struggle with accurate grounding for EgoExo-4D and LEMMA than CharadesEgo due to their lengthy videos and short moments, as shown in Fig. 2. Despite domain differences, we find consistent findings in Table 1, significantly lagging in consistency compared to single-viewpoint performance.

### C.3  THE IMPACT OF REASONING LENGTH

We investigate how the length of the generated reasoning influences optimization stability and performance. Specifically, we categorize reasoning outputs into three groups: short (128 tokens), medium (256 tokens), and long (512 tokens), and train each configuration for 2k steps. Note that we chose the medium-length reasoning in View-GRPO. As shown in the Fig. 8, the medium-length reasoning achieves the most stable rewards and balanced accuracy, while the short and long variants exhibit distinct issues: (1) Short reasoning tends to produce

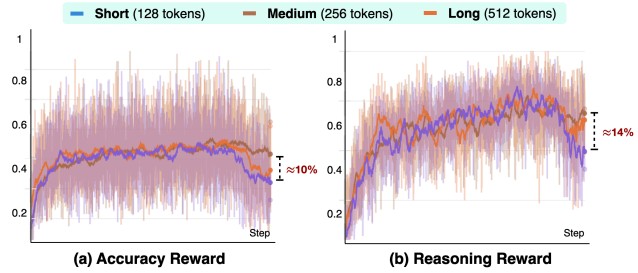

Figure 8: **The rewards across different reasoning lengths.** The medium-length reasoning shows the most stable rewards and balanced accuracy, suggesting that reasoning length critically affects optimization stability.

insufficient context, resulting in low accuracy and unstable optimization. (2) (2) While long reasoning achieves high reasoning rewards, it often leads to lower accuracy, indicating over-exploration or hallucination. This also exposes a limitation of LLM-based judges, where higher rewards do not necessarily reflect factual correctness, as the judge may prioritize verbosity or perceived reasoning depth over actual accuracy. We also experimented with longer reasoning length (1024 tokens <), but did not observe further improvements; in fact, training became less stable and prone to over-exploration. These findings indicate that reasoning length critically affects optimization stability and that excessively short or long reasoning can be detrimental.

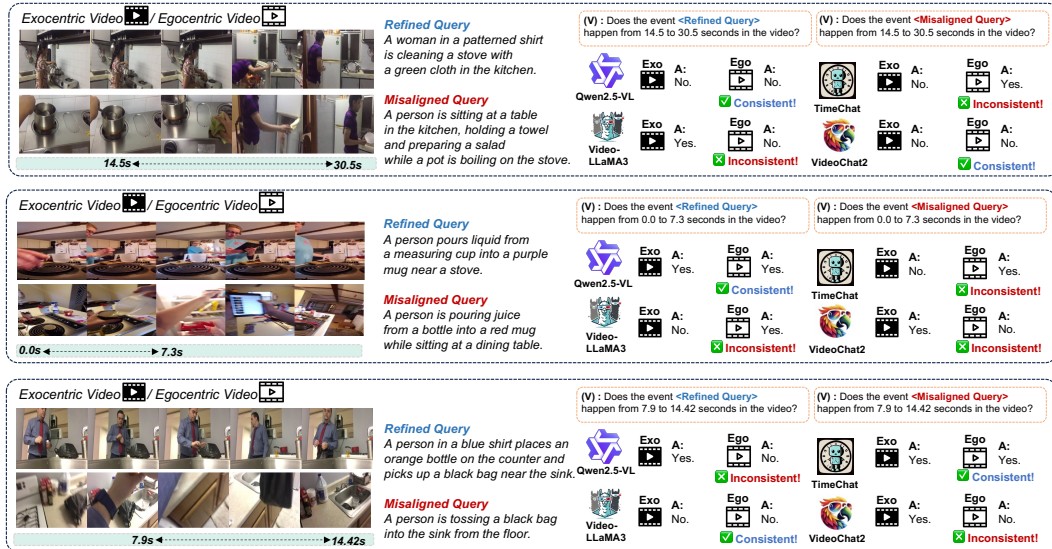

Figure 9: **Examples of test clips and model responses for temporal verification.**. Each row is from CharadesEgo and shows an exocentric–egocentric pair, a *Refined Query* (positive) and a *Misaligned Query* (negative), and per-view answers from each model. We mark consistency (✓) and inconsistency (✗) cases.

## C.4 VIEW-GRPO WITH DIFFERENT BACKBONES

To confirm the effectiveness of View-GRPO with different backbones (*i.e.*, beyond Qwen2.5-VL series), we utilize InternVL-3.5 (Wang et al., 2025a) as our backbone. As shown in Table 6, View-GRPO significantly improves the backbone, demonstrating its robustness.

## C.5 PERFORMANCE OF VIEW-GRPO ON OTHER BENCHMARKS

We test our model (*i.e.*, Qwen2.5-VL-7B trained with View-GRPO) on other benchmarks, Video-MME (Fu et al., 2024) and TVGBench (Wang et al., 2025b). The results in Tab. 7 show that our method consistently outperforms other baselines. We believe that the strengthened temporal reasoning and cross-view consistency through View-GRPO yield synergistic benefits

Table 7: **Performance on Video-MME and TVGBench.**

| Methods | Video-MME | TVGBench | | |
|---|---|---|---|---|
| | w/o subs | R1@0.3 | R1@0.5 | R1@0.7 |
| TimeChat | 30.2 | 22.4 | 11.9 | 5.3 |
| TimeSuite | 46.3 | 31.1 | 18.0 | 8.9 |
| Qwen2.5-VL | 61.1 | 28.1 | 19.5 | 10.5 |
| **View-GRPO** | **69.7** | **42.0** | **25.0** | **13.9** |

and robustness that extend beyond EgoExo-Con, contributing to the observed improvements on both QA and grounding benchmarks.

# D ADDITIONAL VISUALIZATION

We further provide model responses on EgoExo-Con across tasks. Fig. 9 shows model responses for temporal verification from CharadesEgo, and Fig. 10 illustrates grounding predictions from LEMMA and EgoExo-4D. Note that we do not utilize misaligned queries for temporal grounding.

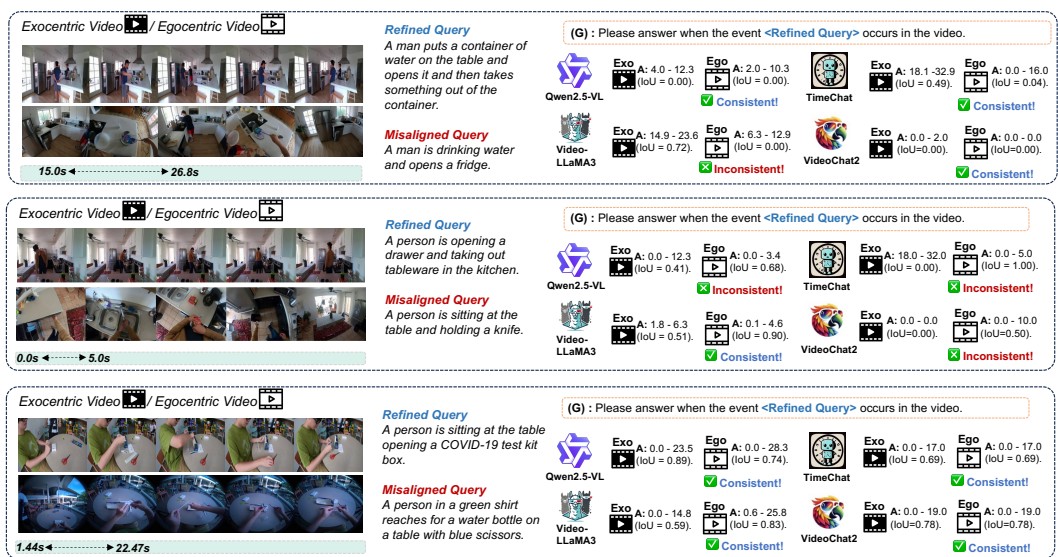

Figure 10: **Examples of test clips and model responses for temporal grounding**. The videos in the top two rows are from LEMMA, and the bottom row shows EgoExo4D videos. Each row presents an exocentric–egocentric pair with a *Refined Query*; models output per-view time spans (with IoU to ground truth when available), and we indicate cross-view consistency (✓) vs. inconsistency (✗).

Refining and Generating Misaligned Captions Using Multi-View Video Understanding.
You are given:
1) A natural language caption describing a sequence of human activities in a video,
2) Four key frames from the egocentric (first-person) view,
3) Four key frames from the exocentric (third-person) view.

Your task is to generate **two outputs**:

**Step 1: Refined Caption**
- Refine the original caption into a **detailed, precise, and coherent single sentence** based only on information clearly observable from **both** the egocentric and exocentric views.
- Replace vague expressions (e.g., "someone," "something") with explicit descriptions (e.g., "a person," "a cup") if reliably identified.
- Only contain mainly interacted object with humans. Do not include background information or objects that might not be clearly visible.
- Maintain temporal coherence and a natural sentence structure.
- Do not hallucinate or invent unseen actions, objects, or attributes.
- Exclude details visible only in one viewpoint but not reliably inferred from the other.

**Step 2: Misaligned Caption**
- Create a **misaligned refined caption** by introducing slight hallucination or inconsistency:
  - Fabricate or modify actions, objects, or attributes not visible in the frames.
  - Break temporal order or describe objects incorrectly.
  - Make the caption fluent and natural but subtly wrong compared to the video evidence.

---

**Example:**
Let us suppose you got the input caption as "A person is standing in a room with two cameras on tripods and a table with a rapid test kit and a timer.", and given frames support this.
However, we should focus on a main character with interacted objects. There we can generate a more concise caption like "A person is sitting at a table with a test kit."
You should follow the below example from the given input:

Input: A person is standing in a room with two cameras on tripods and a table with a rapid test kit and a timer.

Output:
{
  "refined_query": "A person is sitting at a table with a test kit.",
  "misaligned_query": "A person is standing and reading a book."
}

Figure 11: **Prompt for refined and misaligned queries.**

You are an expert evaluator for reasoning quality in video understanding tasks.

You are given:
1. A **ground-truth reasoning** describing the video content.
2. A **candidate reasoning** generated by a model.

Your task:
- Judge how similar the candidate reasoning is to the ground-truth reasoning in terms of semantic content, temporal logic, and event consistency.
- Focus only on factual and reasoning overlap, not wording or style.
- Provide a single float score between **0.0 and 1.0**:
  - 0.0 means completely unrelated or incorrect reasoning.
  - 1.0 means perfectly aligned reasoning.
  - Values in between reflect partial similarity.

Ground-truth reasoning:
"{ground_truth}"

Candidate reasoning:
"{candidate}"

Return only the float score, such as 1.0 or 0.5, with no explanation.

Output:

Figure 12: **Prompt for the reasoning reward function in View-GRPO.**

```
You are an expert multimodal AI assistant specializing in video understanding and temporal reasoning. Your mission is to provide
a thoughtful and comprehensive analysis that determines whether a queried event occurs within a specified video interval.
Your primary task is to carefully analyze the given interval in the video and decide if the described event takes place during
that time window. Output both your reasoning and the final binary answer: "yes" or "no."
### Instructions:

For every (video, query, interval) triplet provided:
    — Thoroughly analyze the video, focusing only on observable cues within the specified interval.
    — Start your reasoning by describing the overall scene: participants, setting, and activities.
    — Examine the interval step by step: describe the sequence of visual cues, motions, or events relevant to the query.
    — Decide whether the event described in the query occurs within the given interval. Only rely on evidence observable in the
      video. If the event is not visible or insufficient cues exist, answer "no."

Strictly follow the output format below.
### Key rules:

    — The output must contain a <think></think> block with detailed reasoning first.
    — Then provide the final binary decision ("yes" or "no") inside an <answer></answer> block.
    — Do not mention "frames" in reasoning; timestamps should always be in seconds.
    — Assume the query is valid and focus only on verifying its occurrence in the interval.

### Example

**Query:** Does the person sweep the floor from 15.0 to 30.0 seconds?
**Interval:** [15.0, 30.0]

Output: <think>First, I analyze the overall context. The video shows a person cleaning a room.
Between 0—15 seconds, the person dusts a bookshelf.
From 15.0 seconds onward, the person picks up a broom and begins sweeping the floor.
The sweeping motion continues clearly until the 30—second mark, with visible back—and—forth arm motions consistent with sweeping.
Therefore, the event "sweeping the floor" indeed occurs within the interval [15.0, 30.0].</think>
<answer>yes</answer>
```

Figure 13: **Prompt for generating temporal reasoning for temporal verification.**

You are an expert multimodal AI assistant specializing in video understanding and temporal reasoning. Your mission is providing a thoughtful and comprehensive analysis, which specify a timestamp of event in given queries. Your analysis and reasoning about events in video streams must be logically valid and universally understandable.

Your primary task is to provide a clear, thorough analysis for each query that identifies the timestamp(s) in the video that best correspond to the described event(s), relying only on cues directly observable in the video. Output precise timestamps and evidence-based reasoning grounded in visible content. Do not evaluate whether the query is correct or whether the event truly occurs; assume the query is valid.

## Instructions
        For every (video, query, timestamp) triplet provided, and the query contains a certain event, which is our interest:

        1. **Thoroughly analyze the video** (via available frames, transcript, or shotlist as appropriate) from a universal perspective, attending only to what is visually, temporally, or contextually observable (not external knowledge or assumptions).
        2. **Start your reasoning by describing the overall scene:** Briefly state the participants, setting, and main activity to specify the event in a given query.
        3. **Decompose the event temporally:** Enumerate all visual cues, transitions, and sub-events leading up to and including the action or moment specified in the query. Assign timestamps (to the extent they can be inferred) to each relevant cue or sub-step.
        4. Your explanation should not depend on subjective or hidden factors; only refer to features and temporal cues directly evident from the video itself.

        **Key rules:**
        – Strictly follow the field order ("reasoning" before "timestamp(s)"), and do not reveal a timestamp before all reasoning is complete.
        – Please provide an output in a string format.
        – A <think></think> block with your detailed, step-by-step, reasoning, which is relevant to given video content. The reasoning should start with analyzing the whole video content and go deeper for specifying a specific video moment.
        – A <answer></answer> block with the final predicted timestamp as `[start_seconds, end_seconds]`. The timestamps should be integers or floats.

        Here is the example:
        Let us suppose that given video frames show a person is cleaning and doing chores.
        Query: Localize the event 'the woman and man are playing piano while sitting on the sidewalk'
        Timestamp: [15.0, 30.0]

        Output: "<think>First, I will establish the overall context. The user is asking for the time period when the person is sweeping the floor. \n1.From the beginning of the video (0 seconds) until the end ({duration} seconds), a person performing a sequence of cleaning chores in a room.\n2.Now, I will pinpoint the 'sweeping the floor' action by analyzing the sequence of events.\n3. From the beginning of the video (0:00), the person is dusting a bookshelf.\n4. At 00:15, the person puts down the duster, picks up a broom, and begins sweeping. This sweeping motion continues until the 30-second mark.\n5. After that, the person sets the broom aside and grabs a dustpan to collect the pile of debris.\n6. This clear sequence (dusting → sweeping → using a dustpan) confirms that the target action is precisely framed. Therefore, the precise time period is [15 to 30].</think> <answer>[15, 30]</answer>"

        **Do not mention 'frame' in the answer as sampled frame number can be varied. Make sure write a timestamp in seconds.**

Figure 14: **Prompt for generating temporal reasoning for temporal grounding.**

