# OpenReview forum: "EgoExo-Con: Exploring View-Invariant Video Temporal Understanding"
_ICLR.cc/2026/Conference — Submitted to ICLR 2026_

### Official Review · Reviewer_q6Jj · 2025-10-31

**Soundness:** 3
**Presentation:** 3
**Contribution:** 3
**Rating:** 6
**Confidence:** 3

**Summary:**

This paper investigates the ability of Video-LLMs to maintain consistent temporal understanding of events captured from different viewpoints. The authors introduce EgoExo-Con, a new benchmark consisting of synchronized egocentric and exocentric video pairs with refined natural language queries for temporal verification and grounding tasks. Through extensive experiments, they reveal that current SOTA models struggle significantly with cross-view consistency, often achieving scores that are much lower than their single-view performance. The paper also demonstrates that naively fine-tuning models on a mix of both viewpoints is insufficient and can even degrade performance. To address these shortcomings, the authors propose View-GRPO, a reinforcement learning framework that encourages models to develop view-specific reasoning chains while aligning their final temporal predictions, showing improved consistency over standard fine-tuning methods.

**Strengths:**

- EgoExo-Con benchmark is a significant contribution. The authors have been sourcing data from diverse datasets and performing careful filtering and human-backed refinement of queries to ensure they are unambiguous and visible from both perspectives.
- The paper provides a comprehensive evaluation of a wide range of Video-LLMs. The key finding that cross-view consistency scores are "barely over half their single-view performance" and that naive multi-view training fails to improve consistency, is a crucial insight that highlights a fundamental weakness in current architectures and training paradigms.
- The proposed View-GRPO method offers a promising direction for improving consistency.

**Weaknesses:**

- The effectiveness of the proposed View-GRPO method is demonstrated only on the Qwen2.5-VL model family. While the results are positive, application to wider range of models would be necessary to make a stronger claim about the generalizability and robustness of the approach.
- Authors showed that the choice of judge model impacts performance. However, the issue of reliability and bias in judge is left for future work. Given its central role in the method's success, a more thorough analysis or ablation study (e.g., removing the reasoning reward) is wanted.
- The View-GRPO is an incremental approach built on top of GRPO. The novelty lies in its application to the cross-view consistency problem and the design of the reward function.

**Questions:**

This paper presents a valuable and timely contribution by defining and benchmarking the problem of view-invariant temporal understanding. The EgoExo-Con dataset and the accompanying analysis are strong points that will benefit the community.
The proposed View-GRPO method, while seems promising, has weaknesses in presentation.

---

> ### Author Response · Authors · 2025-11-21
> **Official Comments by Authors**
>
> ### **[W1] View-GRPO with a different backbone**
> Thanks for the suggestion. To address your concern, we train InternVL-3.5-8B [1] as a backbone and evaluate it on EgoExo-Con. As shown in the table, View-GRPO significantly improves the backbone, demonstrating its robustness. We will include this result in the paper.
>
> | **Methods**         | V-Exo | V-Ego | V-ExoEgo | G-Exo | G-Ego | G-ExoEgo |
> |---------------------|:-----:|:-----:|:--------:|:-----:|:-----:|:--------:|
> | InternVL3.5-8B      | 64.4  | 64.7  | 50.7     | 12.8  | 6.7   | 3.0      |
> | **+ View-GRPO**     | **73.1** | **74.4** | **62.4** | **20.5** | **16.8** | **10.6** |
>
> ### **[W2] Further analyses on the impact of LLM judges**
> We appreciate the reviewer for raising this insightful point. Indeed, the design of the reasoning component is central to our framework. To further analyze this, we investigate how the length of the generated reasoning influences optimization stability and performance. Specifically, we categorize reasoning outputs into three groups: short (128 tokens), medium (256 tokens), and long (512 tokens), and train each for 2k steps.
>
> As shown in Fig. 8 in Appendix C.3, the medium-length reasoning achieves the most stable rewards and balanced accuracy, while the short and long variants exhibit distinct issues: (1) Short reasoning tends to produce insufficient context, resulting in low accuracy and unstable optimization. (2) While long reasoning achieves high reasoning rewards, it often leads to lower accuracy, indicating over-exploration or hallucination. This also exposes a limitation of LLM-based judges, where higher rewards do not necessarily reflect factual correctness, as the judge may prioritize verbosity or perceived reasoning depth over actual accuracy.
>
> We also experimented with longer reasoning length (1024 tokens), but did not observe further improvements; in fact, training became less stable and prone to over-exploration. These findings indicate that reasoning length critically affects optimization stability and that excessively short or long reasoning can be detrimental. We will include this discussion in Appendix in the final paper for completeness.
>
> ### **[W3] Contribution of View-GRPO**
> Thanks for the comment. To our best knowledge, View-GRPO is the first to successfully adapt GRPO-style optimization to improve cross-view temporal prediction consistency. By proposing View-GRPO and View30K, we take an initial step toward encouraging view-specific reasoning while aligning final conclusions across viewpoints. Although the mechanism is simple, our approach proves highly effective, particularly in improving view-invariant temporal understanding of modern backbones such as Qwen2.5-VL.
>
> > References
>
> [1] InternVL3.5: Advancing Open-Source Multimodal Models in Versatility, Reasoning, and Efficiency, arxiv 2025

---

> > ### Comment · Reviewer_q6Jj · 2025-11-28
> > **Response to Authors**
> >
> > Thank you for your response. I will keep my original recommendation.

---

### Official Review · Reviewer_EdFk · 2025-10-31

**Soundness:** 3
**Presentation:** 3
**Contribution:** 3
**Rating:** 6
**Confidence:** 5

**Summary:**

This paper studies the cross-view consistency in predictions of video models.

* First, the authors propose a benchmark (EgoExo-Con) of 491 synchronised videos and 3,178 temporal-bounded queries. While video models perform well in individual views, their predictions are not consistent across views.
* It is shown that naive SFT on a mix of ego-exo video data does not improve this consistency likely due to conflicting priors.
* Finally, the authors propose View-GRPO, an RL based approch (along with a training dataset View30K) to enhance temporal reasoning while encouraging view-invariant video comprehension. This outperforms SFT and standard GRPO. View-GRPO includes three reward signals:
    1. Format reward (for answer structuring)
    2. Accuracy: irrespective of view, the final answer consistency is encouraged by this reward
    3. Reasoning reward: the model is encouraged to output reasoning traces similar to those of GPT5 - this balances cross-view temporal reasoning as well as certain view-specific details.

**Strengths:**

1. **Useful problem and benchmark.** The proposed problem on cross-view consistency is interesting and has several practical applications like learning from human demonstrations. The ExoEgo-Con benchmark, although not as big in size, is thoroughly constructed and is useful in measuring this consistency.

2. **Strong baseline.** The proposed View-GRPO is simple, well-formulated and shows strong performance on the proposed benchmark.

3. **Presentation.** The paper is very well written and was easy to understand.

**Weaknesses:**

1. Instead of enforcing cross-view consistency only through the final answer, the View-GRPO method relies on matching reasoning traces for cross-view consistency. This means that the consistency is enforced via the language space and not directly via visual correspondence. In some narrow cases this can be a limitation for examples where it is hard to describe certain visual elements in language.

2. There is still a lot of room for improvement in terms of consistency scores (Tab 3). While this highlights strength of the benchmark, it could strengthen the paper if some discussion is included on what the authors project as potential ways to improve performance (more ego data in pre-training? entirely new algorithms/architectures?).

3. (Minor) There is some overlap with prior work [1] in terms of problem formulation. However, it is not published and can be considered concurrent. Nevertheless, it may be nice to include a deeper discussion on the differences.

4. Some discussion on whether the proposed ViewGRPO affects performance on other standard video benchmarks is desirable. I am not sure if it is LoRA or full finetuning. In case of latter, such evaluation is much needed.

[1] EgoExoBench: A Benchmark for First- and Third-person View Video Understanding in MLLMs
Yuping He, Yifei Huang, Guo Chen, Baoqi Pei, Jilan Xu, Tong Lu, Jiangmiao Pang

**Questions:**

1. Audio is view-invariant. Have you considered including audio in the mix and using Qwen-Omni like models?

2. Is there a way to automatically quantify if a given query for a video is exo-friendly, ego-friendly or both-friendly? For example, "a person laughing" is likely only exo-friendly and "a person holding something tight" is ego-friendly (clearly visible in ego view but perhaps not visible in exo view).

3. Another type of consistency could be to pass both ego and exo views together and ask if they depict the same action or ask to match coresspondences. Have you considered something like that?

---

> ### Author Response · Authors · 2025-11-21
> **Official Comments by Authors**
>
> ### **[W1] Other factors for cross-view consistency**
> Thanks for making this insightful point. We agree that enforcing consistency in the language space may be a limitation for some examples. This limitation seems to reflect a broader challenge in current Video-LLMs, which still rely on language-space reasoning traces for evaluation and feedback. We will explore richer multimodal consistency signals to strengthen viewpoint-invariant evaluation in the future.
>
> ### **[W2] Further directions for improvement**
> We appreciate the constructive comment. **From a data perspective**, while large-scale ego–exo pre-training could eventually yield more view-invariant representations, collecting comprehensively synchronized multi-view data at scale is costly and often impractical. [1] shows that egocentric-style data can be synthesized from exocentric videos by cropping or reprojecting hand–object regions. Such augmentations could help approximate cross-view signals without requiring paired capture.  **From an architecture perspective**, current Video-LLM training paradigms process videos independently, making it difficult for models to perceive and compare viewpoints jointly, i.e. of separate videos. Architectures that enable simultaneous multi-video processing or explicit cross-view alignment signals could substantially improve consistency. **From a learning perspective**, prediction inconsistencies highlight shortcuts in the learning of view-specific biases. We believe that explicitly reinforcing rationale and temporal reasoning could benefit consistency, as shown via View-GRPO. We will enrich such analyses in the final version.
>
> ### **[W3] Comparison with EgoExoBench**
> Thanks for the kind suggestion. We mentioned EgoExoBench in Section 2. It shares our broad goal of evaluating ego–exo video understanding in Video-LLMs. However, its focus is on cross-view prediction correctness, not prediction consistency. In contrast, our EgoExo-Con targets temporal understanding and consistency across synchronized viewpoints.
>
> Additionally, EgoExoBench highlights the viewpoint gap and offers a benchmark, but no concrete methodology. In contrast, our work reveals the consistency gap and proposes a solution: View-GRPO with our curated View30K dataset, to enhance view-invariant temporal understanding. We will include this discussion in the final paper.
>
> ### **[W4] Performance of View-GRPO on other benchmarks**
> Apologies for missing the details. View-GRPO freezes the visual encoder and updates only the parameters of the LLM via LoRA during training. We test View-GRPO on the other benchmarks, Video-MME [2] and TVGBench [3], and confirm the effectiveness of our method. See the detailed answer for reviewer **88A9** ([W4]).
>
> ### **[Q1] Including audio modalities**
> We appreciate the suggestion on audio and Qwen-omni. The current work emphasizes visual appearance variation across views, and the queries are designed to study cross-view visual correspondence. Incorporating audio will be valuable to help identify some visual events (e.g., knock something), and we will explore integration of additional modalities in the future.
>
> ### **[Q2] Determining either ego or exo and both friendly data**
> Thanks for the interesting question. We believe that powerful LLMs (if properly prompted) could be able to automatically identify some queries that are friendly under a certain viewpoint due to their rich world knowledge. However, since we set rigorous criteria and ensure all queries are visible in both views, we believe that such prior knowledge would rarely help on EgoExo-Con, and sometimes may produce misleading results. We will explore this further to reduce the need for a labor-intensive annotation process.
>
> ### **[Q3] Consistency evaluation in a different way**
> Thanks for the suggestion. We initially considered the way you mentioned to feed both data of viewpoints into a model and give queries. However, current Video-LLMs are trained and evaluated by processing videos independently so this will require architectural changes to consider both viewpoints. We will explore this for future work.
>
> > References
>
> [1] Unlocking Exocentric Video-Language Data for Egocentric Video Representation Learning, arxiv 2024
>
> [2] Video-mme: The first-ever comprehensive evaluation benchmark of multi-modal llms in video analysis, CVPR 2025
>
> [3] Time-R1: Post-Training Large Vision Language Model for Temporal Video Grounding, NeurIPS 2025

---

> > ### Comment · Reviewer_EdFk · 2025-11-24
> > **Satisfied with the response**
> >
> > I thank the authors for a detailed response to my initial review.
> >
> > I acknowledge the authors' efforts in sketching out several directions of improvement (from data/model/learning perspective as well as including audio).
> >
> > About performance on other benchmarks, it is interesting to see that ViewGRPO does better than the base model (Qwen2.5VL) on standard benchmarks like VideoMME. I would urge the authors to include this experiment (and perhaps evaluation on other standard benchmarks as well) in the paper. It is a strong result. Plus, do you have intuition of why it helps on these benchmarks? Is there a mix of ego-exo data in these benchmarks?
> >
> > Overall, I am satisfied by the author response.

---

> ### Author Response · Authors · 2025-11-25
> **Official Comments by Authors**
>
> We are grateful for the reviewer's thoughtful follow-up discussion.
>
> TVGBench includes videos from both exocentric and egocentric datasets. Although Video-MME does not explicitly contain ego–exo videos, both benchmarks likely demand strong generalization capabilities due to their diverse video sources. We believe that the strengthened temporal reasoning and cross-view consistency through View-GRPO yield synergistic benefits and robustness that extend beyond EgoExo-Con, contributing to the observed improvements on both QA and grounding benchmarks.
>
> As you suggested, we have included the results in Table 7 and will also include this discussion in Appendix C.5 in the paper. Thank you once again for your time and thorough review.

---

### Official Review · Reviewer_88A9 · 2025-11-01

**Soundness:** 2
**Presentation:** 3
**Contribution:** 2
**Rating:** 2
**Confidence:** 5

**Summary:**

The paper introduces evaluation benchmark EgoExo-Con to evaluate cross-view video temporal understanding. The authors propose View-GRPO and construct View30K, to explicitly enhance temporal reasoning.

**Strengths:**

1. The paper is clearly written.
2. The proposed EgoExo-Con eval set can be useful for this area of research.
3. The proposed reinforced approach enhanced view-invariant comprehension in video-LLMs.

**Weaknesses:**

1. The proposed evaluation set EgoExo-Con only contains 491 items, which is pretty small and is hard to say weather it is simply finding a hard set for the current Video LLMs.
2. The close-sourced and human performance are reported on a randomly sampled subset, which may be sensitive to sample selection and cannot be fairly compared with open-sourced models and proposed model.
3. In Figure 1 (b), I don't think it is appropriate to expect model understanding "put a knife" from the provided exo video. Because the object of interest is too small. Even for human, it is hard to identify that action. The top example in Figure 3 also makes me confused about how could human even be able to understand the person is opening a cabinet door.
4. The evaluation is pretty limited, only on the proposed set. And compared Table 1 and Table 2, the proposed method is not better than previous models, e.g. TimeChat-VT.

**Questions:**

1. Can you somehow apply your View-GRPO approach with existing model on tasks like Learning Fine-grained View-Invariant Representations from Unpaired Ego-Exo Videos via Temporal Alignment? It is also tackling the similar problem.
2. How many video frames do you take for training? The training time seems too long --"8 xA100 GPUs and requires over 1 day for the 3B model and 2 days for the 7B model".

---

> ### Author Response · Authors · 2025-11-21
> **Official Comments by Authors [1/2]**
>
> ### **[W1] Small scale and hard set**
> Thanks for the comment. We wish to clarify that our goal is not simply to identify hard samples for evaluating Video-LLMs, but to establish a benchmark for fine-grained and consistent temporal understanding across viewpoints. EgoExo-Con enforces strict cross-view synchronization and ensures that query–timestamp pairs are reliably inferable from both views. Such constraints are currently not present in modern benchmarks. While such rigor naturally reduces dataset size, our benchmark (982 videos and 3,178 time-bounded queries) remains comparable in scale to other ego-exo or fine-grained temporal reasoning benchmarks, e.g., EgoExoLearn: 747 videos (432 ego and 315 exo videos), Video-MME: 900 videos.
>
> ### **[W2] The subset for human and closed-source models**
> Thanks for highlighting this concern. We evaluate the open-source models on the subset and the closed-source models on the full dataset. As shown in the results below, the performance gap between evaluating open-source models on the subset versus the full set is minimal. Similarly, the closed-source models achieve comparable performance on the full set as they do on the subset. We will update the paper on the full test set results.
>
> #### Subset
>
> | **Methods**     | V-Exo | V-Ego | V-ExoEgo | G-Exo | G-Ego | G-ExoEgo |
> |-----------------|:-----:|:-----:|:--------:|:-----:|:-----:|:--------:|
> | GPT-5                 | 60.5  | 61.3  | 52.5     | 34.5  | 32.8  | 20.1     |
> | Gemini-2.5 Flash       | 70.4  | 70.1  | 52.3     | 42.0  | 45.9  | 20.8     |
> | TimeChat              | 46.6  | 47.2  | 24.2     | 19.7  | 20.3  | 14.4     |
> | TimeChat-VT          | 62.6  | 60.1  | 41.1     | 27.1  | 25.6  | 15.7     |
>
> #### Fullset
>
> | **Methods**     | V-Exo | V-Ego | V-ExoEgo | G-Exo | G-Ego | G-ExoEgo |
> |-----------------|:-----:|:-----:|:--------:|:-----:|:-----:|:--------:|
> | GPT-5           | 59.8  | 61.0  | 52.2     | 33.8  | 32.7  | 19.9     |
> | Gemini-2.5 Flash    | 70.8  | 70.4  | 52.6     | 41.7  | 46.3  | 20.3     |
> | TimeChat        | 48.9  | 48.4  | 25.1     | 21.3  | 20.5  | 12.8     |
> | TimeChat-VT     | 62.1  | 61.4  | 42.1     | 27.8  | 26.2  | 16.3     |
>
> ### **[W3] Samples in Figures**
> Thanks for pointing these out. Actually, Fig.1 is designed to illustrate how existing datasets are often not well synchronized or positioned (hence the knife being too small).  The figure exactly highlights the need for manual refinement. Please note that these samples are either discarded or refined to ensure reliable evaluation. For the mentioned query "opening a cabinet door" - this is not shown in the figures (nor should a human be able to identify such an action) because it serves as a negative sample (i.e., misaligned query).  The purpose is to evaluate whether a model can correctly confirm this distractor.
>
> ### **[W4] Limited evaluation and inferior performance**
> Thanks for the comment. View-GRPO is designed to improve temporal prediction consistency of the same event across viewpoints. Our constructed EgoExo-Con is currently the only dataset that supports this task. However, as suggested, we zero-shot test our model (i.e., Qwen2.5-VL-7B trained with View-GRPO) on other benchmarks, Video-MME [1] and TVGBench [2]. The results below have shown that our method consistently outperforms other baselines.
>
> | **Methods** | **Video-MME** |  **TVGBench**|        |        |
> |-------------|---------------|--------------|--------|--------|
> |             | w/o subs      | R1@0.3       | R1@0.5 | R1@0.7 |
> | TimeChat    | 30.2          | 22.4         | 11.9   | 5.3    |
> | TimeSuite   | 46.3          | 31.1         | 18.0   | 8.9    |
> | Qwen2.5-VL  | 61.1          | 28.1         | 19.5   | 10.5   |
> | View-GRPO   | **69.7**          | **42.0**         | **25.0**   | **13.9**   |
>
> We wish to clarify that our proposed method is not reported in Tab.1 and Tab.2. Instead, Tab.1 reports the performance of existing models on EgoExo-Con, and Tab.2 analyzes naive baselines of combining coarsely synchronized data - the inferior results are exactly what motivate our method. Compared to TimeChat-VT and ours, View-GRPO significantly outperforms it in the consistency metrics, V-EgoExo and G-EgoExo, demonstrating its improved consistent, view-invariant temporal understanding.

---

> > ### Author Response · Authors · 2025-11-21
> > **Official Comments by Authors [2/2]**
> >
> > ### **[Q1] Applying GRPO to AE2**
> > We appreciate the reviewer’s insightful suggestion. While AE2 [3] addresses a conceptually related problem, it primarily operates at the visual encoder level, focusing on learning view-aligned representations via contrastive objectives. In contrast, our reinforced approach targets reasoning and response generation in natural languages, which AE2’s architecture does not support. Nonetheless, we agree that integrating AE2’s view-aligned visual representations could complement our framework. We thank the reviewer for this valuable suggestion and plan to explore such integration as part of our future work.
> >
> > ### **[Q2] On the training of View-GRPO**
> > Our training uses 2 FPS (clarified in Sec. 5.3). Training time might heavily depend on the scale of datasets and backbones. We will explore ways to improve efficiency in future work.
> >
> > > References
> >
> > [1] Video-mme: The first-ever comprehensive evaluation benchmark of multi-modal llms in video analysis, CVPR 2025
> >
> > [2] Time-R1: Post-Training Large Vision Language Model for Temporal Video Grounding, NeurIPS 2025
> >
> > [3] Fine-grained View-Invariant Representations from Unpaired Ego-Exo Videos via Temporal Alignment, NeurIPS 2023

---

### Official Review · Reviewer_xFbE · 2025-11-04

**Soundness:** 3
**Presentation:** 3
**Contribution:** 2
**Rating:** 4
**Confidence:** 4

**Summary:**

This paper introduces EgoExo-Con, a benchmark to evaluate the capability of VideoLLMs in understanding temporal consistency across ego- and exocentric viewpoints of the same events. The benchmark focuses on two tasks, including temporal verification (binary QA) and temporal grounding. The authors found that the existing VideoLLMs achieve good single-view performance while showing poor cross-view consistency. To this end, they propose View-GRPO, an RL-based fine-tuning framework, extending Group Relative Policy Optimization with an additional reasoning reward and a curated dataset to enhance cross-view temporal reasoning. Empirical results on EgoExo-Con show that View-GRPO improves consistency over standard SFT and GRPO baselines.

**Strengths:**

**[S1] Writing and motivation**
- The paper is well-written and easy to follow. The motivation is clearly presented.

**[S2] Experiments**
- The paper covers extensive range of experiments, including evaluation both open- and closed-source models, reporting detailed per-subset results (CharadesEgo, LEMMA, EgoExo-4D), and fine-tuning analyses.

**Weaknesses:**

**[W1] Dataset and task clarification**
- EgoExo-Con combines pre-existing datasets, so its domain diversity still depends on those sources. The paper claims “comprehensive” coverage, but 491 pairs is small compared to modern multimodal benchmarks.
- Evaluating temporal consistency across viewpoints is critical in view-invariant video understanding. Traditionally, cross-view temporal consistency has been evaluated through cross-view phase progression or Kendall's $\tau$. Additionally, cross-view frame or clip retrieval may also measure the capability of temporal consistency of the models. Compared to previous evaluation tasks, temporal verification may provide a limited insight, as it measures whether the model identifies an event as the same or not. The necessity of temporal verification should be clearly presented in the paper.
- In sum, the contribution of the introduced benchmark may not be significant.

**[W2] View-GRPO**
- View-GRPO itself seems not inherently specific to cross-view reasoning. In other words, rewards corresponding to each view can be optimized separately, and co-optimization across viewpoints is not guaranteed.

**Minor issues**
- Repeated entries in reference (e.g., Feng et al., 2025a/b and Grauman et al., 2024a/b)
- Typo: L77 their sing-view
- Figure 6 should be moved before Figure 7 (or swap the order of figures)

**Questions:**

- There are substantial performance gaps between closed-source models and open-source models, as shown in Table 1. What makes closed-source models stronger? This is just a question to know the authors' opinion.
- Please see weaknesses

**Details Of Ethics Concerns:**

The benchmark reuses existing datasets with human videos: data licensing or privacy statements are not discussed. Clarification of consent and redistribution terms would strengthen compliance.

---

> ### Author Response · Authors · 2025-11-21
> **Official Comments by Authors**
>
> ### **[W1] Dataset and Task Clarification**
> **Limited dataset domain and scale.** We understand the reviewer's concern. While there is an inherent domain limitation, EgoExo-Con achieves broader domain coverage than any individual source dataset. Importantly, we wish to clarify that EgoExo-Con is not a direct aggregation.  It is curated to enable evaluation of fine-grained, view-invariant temporal understanding, supported by extensive human verification and refinement. Among benchmarks with such strict criteria (e.g., multi-view videos) in the table, EgoExo-Con is competitive in scale. Please note that curation efforts for these benchmarks are more challenging and not comparable to building standard multi-modal benchmarks by querying LLMs.
>
> | **Datasets**   | **# Video** | **Multi-View** | **Synchronized.** | **Timestamp** | **Manual Anno.** |
> |----------------|:-----------:|:--------------:|:---------:|:-------------:|:----------------:|
> | TempCompass  | 421         | ✗             | ✗         | ✗             | ✗                |
> | Video-MME    | 900         | ✗              | ✗         | ✗             | ✔                |
> | EgoExo-Learn  | 747         | ✔             | ✗         | ✔             | ✔                |
> | **EgoExo-Con (ours)** | 982   | ✔          | ✔     | ✔         | ✔            |
>
>
> **The necessity of temporal verification.** We appreciate the reviewer's comment. Cross-view phase progression and Kendall’s $\tau$ indeed offer valuable ways to measure temporal consistency. However, these metrics assume access to and optimization over continuous visual embeddings.  Current Video-LLMs are primarily evaluated through language-based reasoning outputs; most rely on frozen visual encoders, which makes it less straightforward to apply these embedding-based metrics directly in our current setting.
>
> This is precisely why temporal verification is necessary: it evaluates whether a model can (i) understand when an event occurs in untrimmed videos and (ii) correctly reject misaligned (negative) queries that do not match the video. These aspects go beyond representation similarity and directly test whether the model’s reasoning is grounded in the video content.
>
> But we fully agree that adapting Kendall’s $\tau$–style or cross-view progression metrics for Video-LLMs could offer complementary insights. We will explore incorporating such representation-level consistency analyses in future work to enrich the evaluation provided by EgoExo-Con.
>
> **Significance of Contribution.** Thanks for the comment. We made three major contributions to the proposed EgoExo-Con benchmark:
> (1) We are the first to study Video-LLMs towards view-invariant temporal understanding, and thus EgoExo-Con serves as a valuable testbed (the only one so far).
> (2) We reveal that video-LLMs' relatively stable performances across individual views are largely due to view-specific bias cues versus faithful view-invariant reasoning.
> (3) We propose View-GRPO, which shows promising improvements for view-invariant temporal understanding via explicit optimization on view-shared reasoning tracks.
>
> We believe the above contributions collectively make our EgoExo-Con a valuable benchmark towards view-invariant video-LLM development, as also well-acknowledged by the other three reviewers (**88A9:** "EgoExo-Con eval set can be useful for this area of research." **EdFK:** "Useful problem and benchmark."" **q6Jj:** "EgoExo-con is a significant contribution.").
>
> ### **[W2] Co-optimization in View-GRPO**
> View-GRPO addresses this by **promoting view-specific reasoning while aligning the final conclusions across viewpoints, ensuring that consistency emerges through reasoning**. Although we encourage view-specific reasoning, both viewpoints still depict the underlying concepts (e.g., objects and actions), even if they appear differently. This shared semantic structure naturally guides the model toward consistent reasoning across viewpoints and this is verified in our experiments.
>
> ### **[Q1] Performance gap between closed-source and open-source models**
> Closed-source models are inherently difficult to characterize, but we believe that their larger model size and more diverse data and training tasks play crucial roles. For open-source models, particularly general-purpose ones, more tailored training data and methods for temporal understanding tasks are much needed for improvement.
>
> ### **Minor Presentation Issues**
> Thanks for pointing them out, and we will fix them in the revision.
>
> ### **Privacy Statements**
> Thanks for pointing it out. Our videos are sourced from existing public datasets: EgoExo-4D, Charades-Ego, and LEMMA, and we directly inherit the licenses of the original video datasets:
> - EgoExo-4D: https://ego-exo4d-data.org/
> - Charades-Ego: https://prior.allenai.org/projects/charades-ego
> - LEMMA: https://sites.google.com/view/lemma-activity

---

### Author Response · Authors · 2025-11-21
**Official Comments by Authors**

We thank all the reviewers for their valuable time and constructive feedback. We are glad for all the positive comments on our efforts and summarize the additionally requested experiments to address concerns raised by reviewers below:

1. (88A9, EdFk) Evaluation of View-GRPO on other benchmarks. Our method is also effective on VideoQA (Video-MME) and temporal grounding (TVGBench).

2. (q6Jj) View-GRPO with a different backbone (i.e., Qwen2.5-VL-7B -> InternVL3.5-8B). This demonstrates the generalization of our method.

3. (q6Jj) In-depth exploration of LLM judges by various reasoning lengths. This provides further insights into how models behave across reasoning lengths.

4. (88A9) Evaluate open-source models on the 30\% subset and closed-source models on the full set; we found negligible performance variations.

New content from the rebuttal will be updated in our final manuscript (indicated by blue text). We look forward to further discussions with the reviewers.

Best regards,
the authors.

---

### Author Response · Authors · 2025-12-03
**Official Comments by Authors**

Dear AC,

We sincerely appreciate your vauable time to help oversee the review process. During the discussion period, the two positive reviewers `EdFk` and `q6Jj` maintained their positive ratings. Unfortunately, we are unable to receive follow-up responses from the two negative reviewers `xFbE` and `88A9` due to the new policy. Nevertheless, we believe we have thoroughly and carefully addressed all raised concerns.

For instance, reviewer `xFbE` questioned the significance of our benchmark. In response, we clarified that EgoExo-Con is competitive in both domain coverage and scale among human-curated temporal benchmarks, and we emphasized the necessity of temporal verification and our major contributions for evaluating fine-grained, view-invariant temporal understanding. Notably, all other reviewers explicitly acknowledged the value of our benchmark.

Reviewer `88A9` expressed concern that our evaluation was limited to our own set. To address this, we evaluated our model on Video-MME and TVGBench, where it achieved meaningful improvements. Reviewer `EdFk` explicitly recognized this as a strong and noteworthy result.

We respectfully invite you to review our responses. Thank you again for your valuable time and consideration.

Best regards,
the authors.

---

### Meta-Review · Area_Chair_Px5P · 2026-01-05

**Summary:**

This paper aims to investigate and evaluate the consistency of Video Large Language Models (Video-LLMs) in temporal understanding across different camera viewpoints (egocentric vs. exocentric).

The scores given by the four reviewers are 6, 6, 4, and 2, respectively.

**Reviewer Concerns:**

The authors have provided sufficient and convincing responses to the comments of all four reviewers and conducted comprehensive additional experiments. The supplementary experiments show good performance and include tests on other benchmarks, demonstrating the effectiveness of the proposed method. They also evaluated two models, InternVL and Qwen, which show the method’s generalization ability.

In response to the score of 2 given by reviewer 88A9, the authors also addressed the reviewer’s misunderstandings. However, it should be noted that the authors need to provide a more rigorous reliability analysis of the LLM-judge results. In addition, the View-GRPO method still essentially relies on alignment in the language space; there remains substantial room for improvement in the upper bound of cross-view consistency. Moreover, in Fig. 3, some frames are indeed difficult to recognize. If many selected clips in the dataset are similarly hard to identify, then evaluating the model’s ability may be affected not only by temporal understanding. For example, from the first-person view, the model may be able to recognize “cloth”, but from the third-person view, it may be hard to identify it as “cloth”, which would also be judged as an inconsistency between ego and exo views. If the dataset primarily contained very clear and easily recognizable content, it would enable a more accurate assessment of temporal understanding ability.

Meanwhile, I acknowledge the temporal grounding task. However, for the temporal verification task, simply asking the model to answer “yes” or “no” is not an effective way to reflect its temporal understanding ability.

**Reviewer Scores:**

Reviewers who received the additional experiments may increase their scores to some extent. However, certain concerns such as reviewer q6jj’s question regarding the reliability of LLM judges in View-GRPO, have not been fully addressed. Moreover, the authors themselves acknowledge that the method is severely limited by its dependence on the language space.

---

### Decision · Program_Chairs · 2026-01-26

Reject